



# Controls of terrestrial ecosystem nitrogen loss on simulated productivity responses to elevated CO2

Johannes Meyerholt[1,2], Sönke Zaehle[1,3]

[1]Biogeochemical Integration Department, Max Planck Institute for Biogeochemistry, Jena, 07745, Germany
[2]International Max Planck Research School (IMPRS) for Global Biogeochemical Cycles, Jena, 07745, Germany
[3]Michael Stifel Center Jena for Data-driven and Simulation Science, Jena, 07743, Germany

*Correspondence to*: Johannes Meyerholt (jmeyer@bgc-jena.mpg.de)

**Abstract.** The availability of nitrogen is one of the primary nutritional controls on plant growth. Terrestrial
ecosystem nitrogen availability is not only determined by inputs of fixation, deposition, and mineralization, but
also regulated by the rates with which nitrogen is lost through various pathways. Large-scale nitrogen loss rates
have been associated with considerable uncertainty, as process rates and controlling factors of the different loss
pathways have been difficult to characterize in the field. Therefore, the nitrogen loss representations in terrestrial
biosphere models vary substantially, adding to nitrogen cycle-related uncertainty and resulting in varying
predictions of how the biospheric carbon sink will evolve under future scenarios of elevated atmospheric $CO_2$.
Here, we test three published approaches to represent ecosystem level nitrogen loss in a common carbon-
nitrogen terrestrial biosphere model with respect to their impact on projections of the carbon effect of elevated
$CO_2$. We find that despite differences in predicted responses of nitrogen loss rates to biogeochemical and climate
forcing, the variety of nitrogen loss representation between models only leads to small variety in carbon sink
predictions. The nitrogen loss responses are particularly uncertain in the boreal and tropical regions, where plant
growth is strongly nitrogen limited or nitrogen turnover rates are usually high, respectively. This highlights the
need for better resolution of nitrogen loss fluxes through global measurements to inform models.

## 1 Introduction

Given potential detrimental implications of increasing global atmospheric carbon (C) dioxide ($CO_2$)
concentrations, research on the terrestrial compartment of the Earth system has focussed on the potential for the
biosphere to sequester atmospheric $CO_2$ in the future (Bonan, 2008). One important constraint to the terrestrial
vegetation C sequestration potential may be the availability of plant nutrients, primarily nitrogen (N) (Hungate et
al., 2003; Thornton et al., 2007; Gruber and Galloway, 2008; Zaehle et al., 2010a; Fernández-Martinez et al.,
2014).

N enters natural terrestrial ecosystems from the atmosphere through deposition of reactive N species (formed by
lightning or fossil-fuel combustion), as well as through biological N fixation (BNF). N is taken up from the soil
by plants, eventually returned as organic litter and is incorporated into the soil organic matter or becomes
mineralized, meaning that microbial activity converts organic N back to plant nutrients, namely ammonium
($NH_4^+$), which can then be converted to nitrate ($NO_3^-$) during nitrification and be taken up by plants again. This
loop of plant N uptake from the soil and mineralization of organic N can be regarded as the internal N cycle.
However, soil N may also be lost from the ecosystem through gaseous and leaching loss processes (Fig.1).



Thereby, the loop of N entering ecosystems from the atmosphere and leaving them through loss processes and eventually ending up back in the atmosphere can be regarded as the external N cycle.

Numerous studies have demonstrated the effects of including N cycle dynamics when using terrestrial biosphere models (TBMs) to examine terrestrial C cycle responses to elevated atmospheric $CO_2$ concentrations ($eCO_2$) and
climate change (Thornton et al., 2007; Sokolov et al., 2008; Zaehle et al., 2010a; Goll et al., 2012; Wania et al., 2012; Smith et al., 2014). The theoretical effect of previously defined, conceptual C-cycle feedbacks (Cox et al., 2000) may be altered by simulating C-N dependencies: Additional plant C assimilation under increased atmospheric $CO_2$ concentrations may be limited by N availability (Zaehle et al., 2010a). Increased global temperatures may not only increase ecosystem respiration and load the atmosphere with more $CO_2$, but also
stimulate N mineralization and provide more N to support plant C assimilation (Sokolov et al., 2008; Thornton et al., 2009).

The magnitude of N effects on model predictions varies between studies, in part due to differences in how N cycle processes are formulated and included in the models (Zaehle and Dalmonech, 2011). Consequently, TBMs vary in their ability to reproduce the results of $eCO_2$ field experiments (Zaehle et al., 2014; Walker et al., 2015).
To gain understanding of the mechanics underlying this uncertainty, new studies have emerged that assess the influence of variety in the representation of individual N cycle processes in model perturbation experiments (Meyerholt and Zaehle, 2015; Wieder et al., 2015; Meyerholt et al., 2016). However, a similar comparative study of N loss representation in TBMs is still lacking, although differences in N loss representation have been suspected in the past as a driving factor of model divergence in response to perturbation when different TBMs
were compared (Thomas et al., 2013; Zaehle et al., 2014; Walker et al., 2015).

In nature, the pathways for gaseous N loss from ecosystems are manifold (Firestone and Davidson, 1989). The aerobic process of nitrification is the oxidation of $NH_4^+$ to nitrite ($NO_2^-$) and then to $NO_3^-$. It is associated with the emission of nitric oxide (NO) and nitrous oxide ($N_2O$) during the reduction of $NO_2^-$ when oxygen is limiting (Li et al., 2000). Denitrification is carried out by denitrifying soil bacteria that are able to sequentially reduce
oxidized forms of N ($NO_3^- \rightarrow NO_2^- \rightarrow NO \rightarrow N_2O \rightarrow$ dinitrogen ($N_2$)) in the presence of organic matter to produce molecular $N_2$ that is emitted to the atmosphere. Anaerobic conditions are a prerequisite, as oxygen depletion causes oxidized N to act as a substitute electron acceptor to denitrifying bacteria. In the process of denitrification, NO and $N_2O$ may also be emitted, which makes this mechanism particularly climate relevant. Denitrification is considered the most important terrestrial N loss flux with 110 Tg N yr$^{-1}$ estimated for the year
2000 (Bouwman et al., 2013). Volatilization of ammonia ($NH_3$) is of special importance in agriculture and may take place when N from manure or fertilizers cannot react to form $NH_4^+$ in the soil due to alkaline conditions or high soil temperatures and is lost to the atmosphere in its gaseous form.

As for the mechanisms of leaching N loss, the pathway of $NH_4^+$ loss is adsorption to soil particles and the possible leaching of $NH_4^+$ bound in this manner, especially in clay soils (Kowalenko and Cameron, 1976;
Matschonat and Matzner, 1996). In contrast to $NH_4^+$, $NO_3^-$ in soils is prone to leaching losses due to its negative charge and inability to adhere to soil particles. In scenarios of high precipitation or irrigation and high $NO_3^-$ concentrations, $NO_3^-$ can be lost to groundwater through vertical transport and thus cause pollution and reduced ecosystem productivity. This hydrological export of N can also affect dissolved organic N, a flux that has been shown to be of sizable magnitude at some sites (Perakis and Hedin, 2002; Gerber et al., 2010).
Despite such general understanding of the pertinent processes, the reason for the variety of N loss representations in TBMs is the difficulty in properly characterizing N loss fluxes at large spatio-temporal scales in nature, given



the strong variability in space and time of the associated trace gas and water fluxes (Boyer et al., 2006). In addition, the relevant fluxes are also very difficult to measure in the field, especially in the abundance needed to constrain global models. Therefore, modellers need to resort to educated guesses on how to represent this poorly constrained ecosystem flux. Model implementations vary between the application of generic loss terms (e.g.

Thornton and Rosenbloom, 2005; Wang et al., 2010) and the explicit formulation of the constituting loss fluxes by simulating the environmental conditions that are assumed to influence specific loss processes (nitrification, denitrification, leaching, fire; e.g. Xu-Ri and Prentice, 2008; Huang and Gerber, 2015). In the latter case, explicit treatment of gaseous N loss may even enable detailed estimates of the ecosystem emissions of the greenhouse gas $N_2O$ (Zaehle et al., 2011). Between these cases of simplified and complex formulations lie a number of

TBMs that represent N loss fluxes at "intermediate" complexity (Yang et al., 2009; Goll et al., 2012; Smith et al., 2014). Models also differ with respect to the N cycle component from which the respective loss flux is derived. Some models focus their simulation of gaseous N loss processes on soil N turnover, i.e. N mineralization (Thornton et al., 2007; Wang et al., 2010), while others base their calculations on the size of the soil inorganic N pool (Xu-Ri and Prentice, 2008; Zaehle and Friend, 2010). Some TBMs include leaching of dissolved organic N

(DON) directly from the soil organic matter (SOM) N pool (Gerber et al., 2010; Smith et al., 2014). Such heterogeneous representation of ecosystem level N losses is a particular limitation when attempting to estimate the effect of N limitation on terrestrial C sequestration, both at present and under future scenarios.

The aim of this study was to determine the extent to which variation between different N loss algorithms would influence simulated C sequestration responses to $eCO_2$. We expected that the different paradigms of

concentration-based and turnover-based N losses would lead to different predicted N loss magnitudes especially under N stress. With simulated depletion of the inorganic soil N pool under $eCO_2$, modelled N loss should decrease more strongly if it is concentration-based than when it is turnover-based. Assuming largely N-limited vegetation growth and fixed ecosystem N inputs, such differences in N loss rates could lead to different C sequestration responses, demonstrating that the choice of N loss formulation plays a notable role in shaping the

predictions of C-N TBMs in simulated perturbation scenarios.

To examine the impact of different N loss algorithms in a TBM, we added two new alternative N loss modules to the O-CN TBM (Zaehle and Friend, 2010). The original O-CN N loss formulation was in part adopted from Xu-Ri and Prentice (2008) and largely bases gaseous N losses on the concentration of reactive N in the soil compartment. As alternatives, we added two more N loss algorithms that base the largest gaseous N loss flux on

the N mineralization flux from soil organic matter to the soil pool of reactive N, inspired by the formulations presented by Thornton and Rosenbloom (2005) and Wang et al. (2010). Thereby, our selection of N loss formulations encompassed the cases of complex and simplified algorithms mentioned above.

As a simple base scenario, we performed $eCO_2$ experiments with these three O-CN versions that only differed in their N loss algorithms at three temperate test sites that only differed in their climate forcing. This was done to

examine how model functioning was affected in quasi-equilibirum and under $eCO_2$ regarding the calculated N loss fluxes and the effect on C sequestration, and to illustrate the approximate climate sensitivity of these patterns. Next, we performed long-term simulations at a temperate site to gain insight into the centennial-scale effect of the three loss algorithms on the evolution of ecosystem N limitation under $eCO_2$. We then performed $eCO_2$ simulations on a global 2°×2° grid using the three model versions to examine the effects of N loss variety

in different ecosystem types that exhibited inherently different degrees of N limitation and climate regimes.



## 2 Methods

In Sect. 2.1, we describe the three modular N loss algorithms. In Sect. 2.2, we describe the different eCO$_2$ experiments we performed.

### 2.1 O-CN terrestrial biosphere model and nitrogen loss formulations:

As a TBM framework, we used the O-CN model that was fully described in Zaehle and Friend (2010) and Zaehle et al. (2011). We used the standard O-CN N loss formulation ("NL1") and added two more formulations as alternatives ("NL2", "NL3"), based on formulations used in other TBMs (Fig. 1). Here, we describe the three

N loss formulations in detail, supplemented by Appendix B in case of the more complex "NL1" formulation.

### 2.1.1 NL1

The original O-CN representation of ecosystem N loss (Zaehle et al., 2011) follows the representation in the

Lund-Potsdam-Jena-Dynamic-Nitrogen (LPJ-DyN) TBM (Xu-Ri and Prentice, 2008) with additions from the DNDC denitrification and decomposition model (Li et al., 2000). O-CN treats the nitrification and denitrification processes explicitly to determine gaseous losses of NO, NO$_2$, N$_2$O, and N$_2$. In addition, NH$_3$ is subject to volatilization based on soil pH. Leaching of solute NH$_4^+$ and NO$_3^-$ occurs in proportion to the soil water lost by soil drainage. A full description can be found in Appendix B.

The NL1 algorithm determines total ecosystem N loss ($N_L$; Eq. 1) as the sum of gaseous N losses from nitrification ($N_{nit}$), denitrification ($N_{denit}$), and volatilization ($N_{vol}$), as well as leaching ($N_{lea}$):

$$N_L = N_{nit} + N_{denit} + N_{vol} + N_{lea} \qquad . \tag{1}$$

### 2.1.2 NL2

The NL2 approach, inspired by Wang et al. (2010), includes N loss fluxes based on soil N turnover and soil inorganic N concentration. A fixed fraction of the instantaneous net N mineralization flux is lost to the atmosphere to represent gaseous N losses associated with microbial soil processes of nitrification and

denitrification ($N_{L, gas}$; Eq. 2). Gaseous N loss only occurs when net N mineralization is positive:

$$N_{L,gas} = f_{gl} * \max(0, N_{nm}) \qquad , \tag{2}$$

where $f_{gl}$ is the fraction (0.05) of the net N mineralization flux ($N_{nm}$) that is lost in gaseous form. To account for leaching losses ($N_{L, lea}$; Eq. 3), the total soil inorganic N pool ($N_{min}$) is reduced at every time step:

$$N_{L,lea} = f_{ll} * N_{min} \qquad , \tag{3}$$

where $f_{ll}$ is the fraction (0.5) of the soil inorganic N pool lost to leaching. The total ecosystem N loss per ($N_L$; Eq. 4) is then given by the sum of gaseous and leaching losses:

$$N_L = N_{L,gas} + N_{L,lea} \qquad . \tag{4}$$



### 2.1.3 NL3

The N loss formulation NL3, inspired by Thornton and Rosenbloom (2005), describes sequential processes of gaseous loss during net N mineralization, gaseous loss from the soil inorganic N pool, and lastly leaching loss

5     from the remaining soil inorganic N reservoir. Similar to NL2, the net N mineralization flux ($N_{mn}$) is accompanied by a fractional denitrification flux ($N_{L,g1}$; Eq. 5):

$$N_{L,g1} = f_{g1} * \max(0, N_{nm}) \qquad , \qquad (5)$$

10     where $f_{g1}$ is the fraction (0.01) of the net N mineralization flux that is lost in gaseous form. Next, excess inorganic N remaining in the soil after immobilization and plant N uptake ($N_{min}$) is subject to further gaseous loss representing volatilization and denitrification ($N_{L,g2}$; Eq. 6):

$$N_{L,g2} = f_{g2} * N_{min} \qquad , \qquad (6)$$

where $f_{g2}$ is the fraction (0.002) of the soil inorganic N pool lost in gaseous form. Any remaining inorganic N in the soil is then subject to fractional leaching loss in constant proportion ($N_{L,l}$; Eq. 7):

$$N_{L,l} = f_l * \left(1 - f_{g2}\right) * N_{min} \qquad , \qquad (7)$$

where $f_l$ is the fraction (0.1) of soil inorganic N lost to leaching. The total ecosystem N loss per time step ($N_L$; Eq. 8) is then given by the sum of gaseous and leaching losses:

$$N_L = N_{L,g1} + N_{L,g2} + N_{L,l} \qquad . \qquad (8)$$

### 2.2 Forcing and simulation protocol

We conducted three seperate sets of eCO$_2$ simulation experiments, two at the local and one at the global scale.

### 2.2.1 Local simulations I

The first set of local simulations was carried out at a representative temperate forest site ("S1") at the coordinates 6° E, 48° N. We included two more sites ("S2", "S3") that were identical to S1, with the exception that we

35     increased air temperatures by 5 K ("S2") or doubled precipitation ("S3") relative to the climate forcing at S1, thereby creating an ensemble of three "pseudosites". This was done to include the effect of climate variation, but without further confounding the results with influences from e.g. different soil and vegetation histories, keeping the effect of the N loss formulation as isolated as possible. For each model version and each pseudosite, the vegetation and soil C and N pools were spun up to equilibrium over 900 simulation years until the year 1700,

40     using 1901-1930 climate forcing from the merged product of the Climate Research Unit observed climatology and the National Centers for Environmental Prediction reanalysis, CRU-NCEP (Viovy, 2016), as used in Le



Quéré et al. (2016). Model spin-up used atmospheric $CO_2$ concentrations from the year 1860 (Le Quéré et al., 2016), 1850 N deposition rates according to Lamarque et al. (2010), BNF according to the "FOR" approach described by Meyerholt et al. (2016), and vegetation cover from the SYNMAP product (Jung et al., 2006). To limit the driving factors in this theoretical study, we disregarded crop vegetation and subsequently N fertilizer application, as well as land-use change.

The models were then run on a half-hourly time step for 313 simulation years using the climate forcing described above from 1901 onward, ambient $CO_2$ concentrations (after 1860) and transient N deposition (after 1850) to generate unperturbed control model output. For our $eCO_2$ treatment, we added 200 ppm $CO_2$ to ambient concentrations every year from 1950 onward until the simulations ended after the year 2013.

### 2.2.2 Local simulations II

The second set of local simulations was carried out only at the temperate "S1" site from Sect. 2.2.1, with the following modifications to the first set. Instead of 63 years as above, a different $eCO_2$ experiment was conducted over 300 simulation years. After spin-up, atmospheric $CO_2$ concentrations were kept constant at the 1860 level (286 ppm) between 1700 and 1860. Between 1860 and 2006, atmospheric $CO_2$ increased according to ambient concentrations. Next, the 2006 concentration (380 ppm) was kept constant for the following 300 years to create the experiment control runs. For the treatment runs, atmospheric $CO_2$ was set to 580 ppm between the years 2006 and 2306.

Instead of the recorded climate data, randomly selected climate years for the "S1" site were used from the years 1901-1930 throughout all simulations. After the year 2006, atmospheric N deposition rates were kept constant to the 2006 level.

### 2.2.3 Global simulations

Global simulations were carried out on a global grid of 2° x 2° resolution. The ecosystem C and N pools were spun up to equilibrium for 1291 years until 1850, using 1700 vegetation cover (Hurtt et al., 2006), 2000 fertilizer application (Zaehle et al., 2010b), 1850 N deposition rates (Lamarque et al., 2010), 1850 ambient $CO_2$ concentrations, and 1901-1930 global climate data. The climate data was taken from the 5th Phase of the Coupled Model Intercomparison Project (CMIP5) projection of the Institute Pierre Simon Laplace (IPSL) general circulation model IPSL-CM5A-LR (Dufresne et al., 2013), bias-corrected according to the Inter-Sectoral Impact Model Intercomparison Project (ISI-MIP; Hempel et al., 2013). We then performed global simulation runs from 1850 to 2099. None of the forcing was transient after spin-up, except for the atmospheric $CO_2$ concentrations that increased according to the representative concentration pathway 8.5 (RCP8.5; Meinshausen et al., 2011).



## 3 Results

### 3.1 Local simulations I

### 3.1.1 Temperate forest site (S1)

Since N input rates of deposition and fixation were fixed, the total N loss rates predicted by the models for the quasi-equilibrium simulation period (1700-1750) barely differed between pseudosites (Fig. 2a,e,i). At S1, NL3 calculated the lowest fraction of N loss through leaching because leaching was assumed to take place after plant

N uptake and two gaseous loss pathways were accounted for, leaving less N in the inorganic soil pool for leaching (Fig. 2a). Despite the differences in concept and detail of N loss representation, the proportions between gaseous loss and leaching for NL1 and NL2 were predicted to be more even. The simulations that employed the NL1 loss algorithm, however, generated more year-to-year variability than the other simulations, which generally applied across pseudosites. This was because losses in NL1 were mostly dependent on the N

concentration in the inorganic soil pool, which underwent pronounced fluctuations from N plant uptake, losses and inputs, and generally did not accumulate N over substantial periods of time. In contrast, NL2 and NL3 based a large portion of the calculated N losses on the mineralization flux (Fig. 1), which was derived from the pool of soil organic matter (SOM). This pool was far larger in absolute amounts of N than the inorganic pool, making N loss a rather consistent flux in comparison as long as vegetation, i.e. substrate for N mineralization, was present.

Subjected to ten years of $eCO_2$, the simulated ecosystems showed considerable variation in how N loss rates were predicted to evolve, depending on the applied loss algorithm (Fig. 2b). While NL1 and NL3 predicted decreases in total loss rates, NL2 predicted increased gaseous N loss and a slight reduction of leaching rates, resulting in an overall increase of total N loss. The NL1 response was dominated by a decrease of gaseous loss. In the O-CN TBM, $eCO_2$ led to increased plant growth, accompanied by increased plant N demand. When this

demand was met through increased plant N uptake, the soil inorganic N pool became more strongly depleted than it would have under ambient $CO_2$ concentrations. Gaseous N losses in NL1 decreased because they were mostly based on the soil inorganic N concentration.

In contrast, applying the NL2 model under $eCO_2$ at S1 resulted in an increase of gaseous N losses. This was a direct reflection of the exclusive dependence of gaseous loss on net N mineralization in NL2 (Fig. 1). In the O-

CN TBM, the depletion of the soil inorganic N pool under $eCO_2$ led to an increased C:N ratio of SOM, which in turn led to increased N release from the mineralization of organic material to steer SOM C:N back towards a target ratio (Zaehle and Friend, 2010). Thus, when NL2 was applied, the effect on gaseous N loss was an increase. Depletion of soil N also caused a decrease in (concentration-dependent) N leaching, however the net effect (total N loss) was positive (Fig. 2b).

The NL3 algorithm produced reduced total N loss under $eCO_2$ through reduced gaseous loss, albeit at a smaller magnitude than NL1. Although NL3 featured a similar mineralization-dependence of gaseous N loss as NL2, we found that most gaseous loss change in NL3 occured independently of net N mineralization change under $eCO_2$. This meant that any gaseous loss increase that occured with increased net N mineralization was superseded here by the secondary, soil N concentration-based loss pathway that reduced gaseous N loss with soil N depletion

(Fig. 1). As with the NL1 formulation, the $eCO_2$ effect on leaching was negligible for NL3 at S1.



Despite these major model differences in predicted N loss changes under $eCO_2$ at S1, the model differences in predicted C sequestration changes (NPP; Fig. 2c) and ecosystem C accumulation during the experiment (Fig. 2d) were small. All model versions predicted NPP increases around 25% and growth of the total ecosystem C pool between 4 % and 5 %. Interestingly, the NL2 loss model predicted the largest increase in ecosystem C despite also predicting the only increase of ecosystem N loss. In particular, this largest C increase was simulated in the SOM C pool, where subsequently the SOM C:N also increased stronger for NL2 than for the other loss algorithms.

### 3.1.2 S2 site

For the quasi-equilibrium loss partitioning at the S2 pseudosite (Fig. 2e), the most prominent difference compared to the S1 site was that the +5 K change in air temperature caused a smaller leaching portion predicted by the NL1 algorithm. At higher temperatures, soil evaporation increased and soil drainage decreased, which for NL1 led to a reduction of leaching loss as a consequence of the coupling of drainage and leaching in this algorithm (see Appendix B). As this coupling was not applied in the NL2 and NL3 formulations, they predicted barely any effect of higher air temperatures on the partitioning between gaseous loss and leaching.

Notable effects of higher temperatures on the $eCO_2$ responses of N loss rates (Fig. 2f) included a reduction in leaching for NL1 and a stronger gaseous loss reduction for NL3. With the reduced baseline leaching for NL1, the reduction of soil inorganic N concentrations under $eCO_2$ resulted in now notable reduction of leaching loss. For NL3, the gaseous loss reduction was dominated by the secondary, N concentration-based pathway. Given the high temperatures at the S2 site, N mineralization rates were already at a high level and increased under $eCO_2$ at a lower rate than at the S1 site. The rather small fraction (0.01, see Sect. 2.1.3.) of the N mineralization flux that was lost in gaseous form in NL3 did not make for substantial loss through the primary, flux-based pathway. Instead, gaseous losses were reduced stronger than at S1 because more inorganic N was left in the soil to be lost through the secondary loss pathway.

Compared to the S1 site, the temperature increase at S2 resulted in higher NPP responses (over 30%; Fig. 2g) and more ecosystem C accumulation under $eCO_2$ (over 5%; Fig. 2h). Higher temperatures led to higher gross primary productivity (GPP) responses to $eCO_2$ due to the sensitivity of modelled photosynthesis, which subsequently propagated to the NPP and C accumulation responses.

### 3.1.3 S3 site

When precipitation was doubled, the leaching portion of N loss for the NL1 formulation in quasi-equilibrium increased dramatically (Fig. 2i), owing to the dependence of leaching on drainage. Since in this state the total ecosystem N loss was essentially prescribed by the fixed rates of ecosystem N input, the gaseous loss portion was minimized. This was further aided by a decreased nitrification rate in NL1 at S3 due to a decreased aerobic soil fraction (see Appendix B), which reduced the associated gaseous losses. While there was no effect of precipitation increase for NL2, the proportions of loss pathways changed for NL3 towards more leaching and less gaseous loss.

The precipitation increase brought about a number of changes to the sensitivity of N loss under $eCO_2$ (Fig. 2j). For NL1, most of the N loss reduction was now simulated as reduced leaching, a consequence of most NL1 N



loss now occuring as leaching (Fig. 2i). For NL2, the prediction of total N loss change switched from an increase to a decrease, on account of the leaching decrease now being of greater magnitude than the gaseous N loss increase. For NL3, precipitation increase led to strongly increased leaching and gaseous N loss reduction, approximately quadrupling the total N loss reduction compared to S1.

All models predicted NPP responses to $eCO_2$ of approximately 20 % (Fig. 2k) with even predictions between N loss models when precipitation was increased. Model differences were also minimal regarding ecosystem C accumulation (Fig. 2l), however, all models predicted higher accumulation at S3 compared to S1 (approximately 5 %).

**3.2 Local simulations II**

While the previous section dealt with the short term effects of a ten year step-increase in atmospheric $CO_2$ concentrations, the second set of local experiments was designed to investigate long term effects of $eCO_2$ on N limitation of vegetation growth and how sensitive they were to the applied N loss formulation. During the 300 year $eCO_2$ simulation at the temperate "S1" site (300 years step increase from constant control $CO_2$ by 200 ppm), all loss models predicted the total N loss rate to decrease (Fig. 3a). In the long term, the NL1 response was less pronounced than the NL2 and NL3 predictions. Note that the NL2 model produced a positive N loss response early on in the simulation (see also Sect. 3.1), but eventually predicted a negative N loss response close to the NL3 model prediction.

The ratio of total ecosystem N loss and net N mineralization (termed "N cycle openness") was predicted to decline by all models (Fig. 3b). This meant that in all simulations, relative to the control runs, less N was lost from the system compared to new N becoming available from mineralization, i.e. the internal ecosystem N cycle became more "closed". As mentioned in Sect. 3.1.1, net N mineralization generally increased in the O-CN TBM under $eCO_2$. For the models that predicted reduction of N loss at S1 in the shorter term (NL1, NL3; Fig. 2b), reduced N cycle openness was therefore an expected result, notwithstanding the slightly different experimental designs. However, using the NL2 model that calculated ecosystem N loss increase early on resulted in reduced N cycle openness as well, meaning that early N mineralization increased at a higher rate than N loss did.

Model predictions of N cycle openness responses did differ in that NL1 predicted an inital sharper decline relative to the control runs than NL2 and NL3 did (Fig. 3b). This was a consequence of NL1 N loss being largely dependent on the soil inorganic N concentration that declined quickly in response to the step increase of the atmospheric $CO_2$ concentration (Fig. 3a). In contrast, the dependencies of NL2 and NL3 on the N mineralization flux made for a more gradual decline of N loss (and N cycle openness), owing to N mineralization depending on the slower dynamics of the SOM N pool. Over the 300 simulation years, all loss models approached the same approximate absolute magnitude of N cycle openness reduction.

The magnitudes of NPP responses (Fig. 3b) were again largely unaffected by N loss differences, which was expected considering the findings in Sect. 3.1.1 (Fig. 2c). There was a trend of NL1 sustaining a larger NPP response than the other models until simulation year 175, however, this trend dissolved over the following decades of simulation until the end of the experiment.




### 3.3 Global simulations

Having examined the $eCO_2$ effects of N loss variety in detail at a temperate site, we next applied the three loss algorithms in global simulations to observe the dynamics between $eCO_2$, N loss and ecosystem C accumulation for different vegetation types and climate regimes. Notably, only the atmospheric $CO_2$ concentration was varied, whereas climate and N forcing was fixed (see Sect. 2.2.3).

The three model versions were spun-up to quasi-equilibrium for the soil and vegetation C and N pools using 1850 atmospheric $CO_2$ levels according to the RCP8.5 scenario (285 ppm; Meinshausen et al., 2011). Yet, after 138 simulations years and having gradually reached atmospheric $CO_2$ levels of 350 ppm, global N loss rates were still predicted to be similar between the models (Fig. 4a,b,c). This showed that the models did not differ much when responses to gradual, low magnitude $eCO_2$ were calculated. The hotspots of N loss were regions with high density of agricultural land use and, to a lesser extent, the tropical zone with high natural N turnover.

All loss models tended to predict more or less pronounced reductions of total ecosystem N loss in global simulations under $eCO_2$ by the time a 550 ppm atmospheric $CO_2$ concentration was reached, i.e. after 64 more years of simulation (Fig. 4d,e,f). Model predictions differed, however, in the magnitudes of N loss reductions with some notable regional disagreement. Some of the regions for which all models consistently predicted sizable reductions of loss rates were arid parts of the Canadian Prairies, most northern temperate and boreal regions of Russia, as well as regions surrounding the Central Asian deserts, where vegetation cover and baseline N turnover was low, therefore not contributing much to the total global N loss flux in absolute terms. The models also predicted N loss reduction for tropical rainforests, regions with some of the highest global N stocks and turnover rates.

The most notable model differences could be found between the NL1 and NL2 loss models, with NL1 predicting generally large (often greater than 30%) negative N loss responses, and NL2 predicting generally smaller (mostly lower than 20%) negative responses (Fig. 4d,e,g). Approximately, the responses predicted by NL3 could be classified as close to NL2 in the temperate and boreal regions, and close to NL1 in the tropics. The large negative response for NL1 in the boreal regions was a manifestation of the soil N concentration-based N loss fluxes. The boreal regions are usually considered strongly N limited in their vegetation growth, i.e. low on N available in the soil for plant uptake. Therefore, a small absolute decrease in soil N concentration due to plant uptake increase under $eCO_2$ was enough to result in a high relative reduction of total N loss in NL1. This mechanism did not apply as strictly in the NL2 and NL3 models, which resulted in less pronounced responses in the boreal regions. NL2 predicted the smallest decrease in tropical N loss rates because its loss pathways were the least affected by the soil inorganic N concentration decreasing under $eCO_2$, as NL2 used this dependence only to determine leaching.

The model differences regarding N loss responses were most prominent in highly N limited regions (Fig. 4g). Some ecosystems in these regions such as boreal forests are also known to store large amounts of C. This raised the question whether model differences in global N loss responses, including in highly N limited regions, also resulted in appreciable model differences with respect to predicted C accumulation under $eCO_2$.

As the N loss algorithms predicted different rates of N loss change in response to global atmospheric $CO_2$ increase (Fig. 4), there was also disagreement on the amounts of N that would accumulate in ecosystem over the entire global simulation period (1850-2100) when atmospheric $CO_2$ concentrations increased from 285 ppm to 936 ppm (Fig. 5). The predicted N accumulation varied between 1549.7 Tg N (NL1; corresponds to 6.2 Tg N yr⁻



$^{1}$), 1444.5 Tg N (NL2; 5.78 Tg N yr$^{-1}$), and 1444.9 Tg N (NL3; 5.78 Tg N yr$^{-1}$). Ecosystem N accumulation varied in particular in northern temperate and boreal regions, where the NL1 loss model led to the most N accumulation, as well as in tropical ecosystems, where the NL2 loss model led to the most N accumulation. Most of the variety in these predictions of ecosystem N accumulation stemmed from variety in predicted accumulation

in SOM, with the exception that in the tropics, NL1 predicted notably less N storage in vegetation than the other models.

The resulting model disagreement regarding the additional C accumulation under eCO$_2$ was small (Fig. 5). Using the three N loss algorithms led to calculated cumulative terrestrial C uptake of 135.0 Pg C (NL1; corresponds to 0.54 Pg C yr$^{-1}$), 142.3 Pg C (NL2; 0.57 Pg C yr$^{-1}$), and 139.8 Pg C (NL3; 0.56 Pg C yr$^{-1}$) in our eCO$_2$ scenario. In

particular, there were only small model differences simulated for tropical or boreal forests, where the variety in N accumulation was highest.

## 4 Discussion

The variability between N loss algorithms in predicted C accumulation for the RCP8.5 eCO$_2$ scenario between 1850 and 2100 (Fig. 5) was very low compared to the large variability in predictions of different TBMs for a similar scenario (Jones et al., 2013). This indicated that uncertainty in N loss representation was not a major driver for variability in future C sink predictions. This result was obtained in spite of some non-negligible variety in predicted ecosystem N accumulation during the global eCO$_2$ experiment (Fig. 5). The tendency of global

results to indicate that variety in predictions of N loss change (Fig. 4) and N accumulation (Fig. 5) under eCO$_2$ did not have large impact on corresponding predictions of responses in C sequestration was in principle also found at the site level (Fig. 2, 3). We expected that the effect would be limited, because it was shown before that in the O-CN TBM, the magnitude of N loss is about 20 % of the magnitude of plant N uptake (Meyerholt et al., 2016). However, the small margin in C predictions between the N loss algorithms is still remarkable, considering

the different levels of complexity with which the loss fluxes were determined.

The lack of direct connection regarding model variety between N predictions and C predictions appears plausible for the tropical zone, where vegetation growth is typically not considered N-limited. Therefore, N variety in loss rates and accumulation were not expected to affect C predictions strongly. The reasons for the small C effect outside the tropics are, however, less clear. The phenomenon might be explained by the concept of flexible C:N

stoichiometry in organic soil and plant tissues employed in the O-CN TBM (Zaehle and Friend, 2010; Meyerholt and Zaehle, 2015). The buffering capacity of flexible ecosystem C:N ratios could indeed attenuate the variety in N loss responses to eCO$_2$ and render the effect on C accumulation minimal as seen in our results (Fig. 5). However, we found that when we employed fixed ecosystem C:N ratios in both organic soil and vegetation following Meyerholt and Zaehle (2015), predicted C accumulation only became more variable in far northern

latitudes due to variable productivity under strong N limitation, while becoming even more uniform everywhere else (Fig. A1). The variety of global C accumulation predictions was hardly affected (143.9, 135, 138 Pg C for NL1, NL2, NL3; 1850 - 2099). Thus, with the exact mechanisms remaining difficult to discern, we report that within the context of the complex O-CN TBM, variety in N loss algorithms did not lead to great variation in C sink predictions under eCO$_2$, although varying magnitudes of responses in ecosystem N loss rates, mostly

reductions of N loss, were predicted.





Aside from the short term predictions of the NL2 loss algorithm, this reduction effect (Figs. 2, 3, 4) was in line with the expected mechanisms in play for C-N TBMs under $eCO_2$ (Zaehle et al., 2014; Walker et al., 2015), with increased plant N demand causing more N uptake from the soil, leaving less residual N to be lost. However, although denitrification is being considered the dominant mechanism of ecosystem $NO_3^-$ loss (Fang et al., 2015),

most field experiments have not found decreases in $N_2O$ emissions under $eCO_2$. Positive or neutral responses have been common, primarily obtained in temperate or boreal forests or grasslands (Van Groeningen et al., 2011; Dijkstra et al., 2012). It should be noted that none of these field experiments were conducted in the tropics, which may hamper comparisons (Huang and Gerber, 2015), also seeing how a substantial portion of our obtained N loss decrease was observed in these latitudes (Fig. 4). Nevertheless, a number of mechanisms have

been proposed that would explain increased denitrification under $eCO_2$ (Butterbach-Bahl and Dannenmann, 2011), some of them on the level of abstraction with which the N cycle is commonly represented in TBMs. For example, $eCO_2$ could change the plant's water use efficiency through decreased plant transpiration, leading to higher soil water content and a higher likelihood of anaerobic soil conditions that would benefit denitrification. Further, $eCO_2$ could stimulate the rhizodeposition of labile C compounds such as amino acids and sugars,

increasing soil microbial activity and thereby providing benefitial conditions for denitrification through oxygen depletion and provision of organic C. While such processes were in principle represented in the NL1 formulation (see Appendix B), the predicted result was still the reduction of N loss under $eCO_2$, because the trends were stronger determined by the increased plant N uptake and soil mineral N depletion. It might be that these main mechanism proposed by models to reduce N loss under $eCO_2$ just do not apply as generally, especially in very N

rich soils. This leads to the challenge for models to treat the actual size and distribution of global soil N inventories, if modellers hope to draw connections between soil N content and soil N emissions on large spatial scales.

As for other model studies, Huang and Gerber (2015) only found initial reduction of soil $N_2O$ emissions in the tropical biome in a long-term $eCO_2$ simulation using a TBM, followed by a substantial increase and the

responses in other ecosystems being neutral or positive throughout. Aber et al. (2002) demonstrated that in a stand-scale ecosystem model, the N loss (leaching only) response was only negative under simulated $eCO_2$ when the experiment was not confounded by other perturbations such as increased N deposition and climatic change. In that model setting, other perturbations of N input and accelerated N turnover would eventually increase N losses, a result that we also largely obtained globally, when we added increased N deposition and transient

climate to the $eCO_2$ experiment (Fig. A2). While there is still discrepancy between the mechanisms that likely control the N loss response to $eCO_2$ in nature and the mechanisms that shape model responses, the most immediate notion here is that this effect needs to be studied in actual tropical ecosystems that are N rich and will be crucial for the global climate under future change. Also, the above comparisons are limited by differences in the typical durations of field campaigns and model simulation runs. Time scale may well be pivotal here, since

the functioning of the N cycle and its sensitivity to changing climate and biogeochemistry has long been hypothesized to change over longer (decadal and onwards) time scales (Aber et al., 1989; Vitousek and Howarth, 1991; Luo et al., 2004).

Performing a local temperate 300 year $eCO_2$ experiment, we found that initial model differences in N loss rate responses would over time approximately converge to a similar level that remained negative, i.e. N loss

reduction (Fig. 3). The persistent N loss reduction over time meant that using our array of N loss algorithms within the framework of the O-CN TBM never resulted in a prediction of long-term progressive N limitation



(Luo et al., 2004) at the temperate site. Walker et al. (2015) conducted similar local model experiments at temperate sites, but instead of comparing N loss algorithms they compared entire TBMs. They found a variety of N loss responses, from negligible responses to initial reductions that would over time subside and approach zero or even positive responses, which suggested that some TBMs predicted onset of progressive N limitation under a

long term $eCO_2$ regime. The different N loss algorithms applied in these TBMs (some of which were represented in our study) were likely influential in producing the variety of $eCO_2$ responses presented in Walker et al. (2015). However, we showed here that variety in N loss formulations alone was not enough to produce such heterogeneous responses in the long term. Rather, the results produced by different TBMs were confounded by many other assumptions about the C and N cycles that were inherent to each model.

We have shown that the different loss algorithms simulated variable partitioning of N losses between gaseous and leaching losses, both in quasi-equilibrium and for $eCO_2$ responses (Fig. 2). In reality, the partitioning of the two N loss pathways means no less than the difference between predicted air pollution or water pollution if N enriched ecosystems are considered (Aber et al., 1989). Consequently, adequate N cycle representation still mandates a better grasp of this partitioning issue, for which major model discrepancies have been shown before

(Thomas et al., 2013). Some of the model differences were fairly obvious consequences of the respective formulations, such as a very high leaching portion in a high precipitation environment when leaching was a function of drainage (NL1), or the virtual elimination of leaching in the hierarchical structure of NL3 when gaseous N loss was already substantial (Fig. 2). These examples showed that in TBMs that include both gaseous and leaching pathways, there has not been a consensus on the proper partitioning between fluxes that needs to be

reflected. The reason for this is surely the lack of field evidence of simultaneous measurements of both pathways to inform models. Some models not considered here have included leaching only (Gerber et al., 2010; Wania et al., 2012; note that the Gerber et al. formulation was updated in Huang and Gerber, 2015), which further illustrates that across current TBMs, N loss representation and in particular the partitioning between gaseous and leaching losses has not been an issue of priority. Our results that indicated little effect of the N loss

representation on future C sequestration under $eCO_2$ should not encourage continued neglect of this aspect of N cycle modelling (Walker et al., 2015).

Further, deriving gaseous N loss from the N mineralization flux as described e.g. by Thornton and Rosenbloom (2005) and Wang et al. (2010) may be too coarse of an approximation. The "Hole-in-the-pipe" model (Firestone and Davidson, 1989), frequently cited as inspiration, highlighted that the fluxes of nitrification and

denitrification are the important processes from which NO and $N_2O$ emission are derived. At best, mineralization could be considered as a "distal" indicator of nitrification activity, since it aids in providing $NH_4^+$ substrate, one of the controlling factors of nitrification. From the ecosystem modelling perspective, denitrification is usually assumed to be controlled by the presence of $NO_3^-$, organic C and anaerobic conditions, not N mineralization. Likely, the relatively simple algorithms we implemented as NL2 and NL3 were in part the result of the

respective N-cycle model not discriminating between $NO_3^-$ and $NH_4^+$ and rather calculating a generic N species for convenience. This discrimination will likely be necessary if N cycle modelling and N loss algorithms in particular are to be advanced across terrestrial biosphere models.

The above example illustrates that regional scale modelling of N loss fluxes in TBMs is still developing, as current research continues to e.g. investigate the sensitivity of leaching to global change in a TBM (Braakhekke

et al., 2017). Importantly, one of the most climate-relevant aspects of N loss fluxes, $N_2O$ emission, has become the focus of a model intercomparison project aiming to understand past and present $N_2O$ fluxes by utilizing state-




of-the-art models along with available data (Tian et al., 2018). We contend that such efforts to consolidate the representation of ecosystem N loss processes could best be aided by field experiments that investigate N loss rates (and their partitioning between gaseous and leaching components) under perturbation in the regions that we identified as crucial with respect to modelled N loss uncertainty. Our global simulations showed consistent

model disagreement in northern temperate and boreal latitudes on the magnitude of N loss rate changes under $eCO_2$ and a small effect on C sequestration. Vegetation growth in these regions is usually thought to be strongly N limited, which is largely controlled by the ecosystem N budgets, including N loss rates. For future C sink predictions, it might be important to describe how N cycling affects the mobility of the large C reserves in these latitudes that are often considered undersampled with respect to many ecosystem variables. On the other hand,

we found large N loss rate changes in the tropics, but also faced a lack of appropriate field experiments to evaluate these results. While tropical vegetation growth is usually not considered N limited, the flipside is that many of these regions are N rich, with large potential of water eutrophication or the outgassing of NO and $N_2O$, all environmental issues of note where new experiments are urgently needed to inform models.

### Code availability

The used model code is available from the authors upon request.

### Author contribution

JM and SZ designed the study. JM and SZ set up the simulations. JM ran the local experiments, SZ ran the global experiments. JM and SZ analyzed the results. JM wrote the manuscript.

### Competing interests

SZ is an associate editor for Biogeosciences.

### Acknowledgements

This work was supported by Microsoft Research through its PhD Scholarship Programme and the European Research Council (ERC) under the European Union's Horizon 2020 research and innovation programme (QUINCY; grant no. 647204).





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





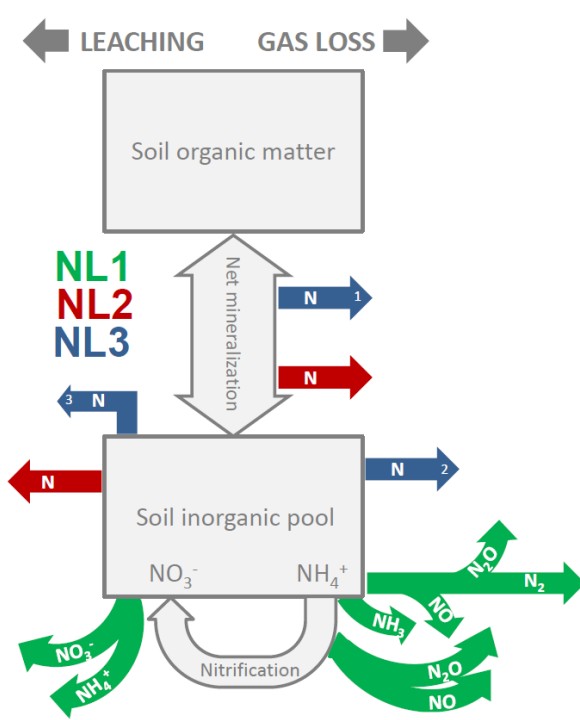

5    **Figure 1: Schematic representation of the three employed nitrogen (N) loss algorithms.** "NL1" is the original algorithm used in the O-CN model, "NL2" and "NL3" were added for the purpose of this study. Numbers in the blue arrows indicate the sequential nature of the "NL3" approach. $N_2$ = dinitrogen; NO = nitric oxide; $N_2O$ = nitrous oxide; $NH_3$ = ammonia; $NO_3^-$ = nitrate; $NH_4^+$ = ammonium.



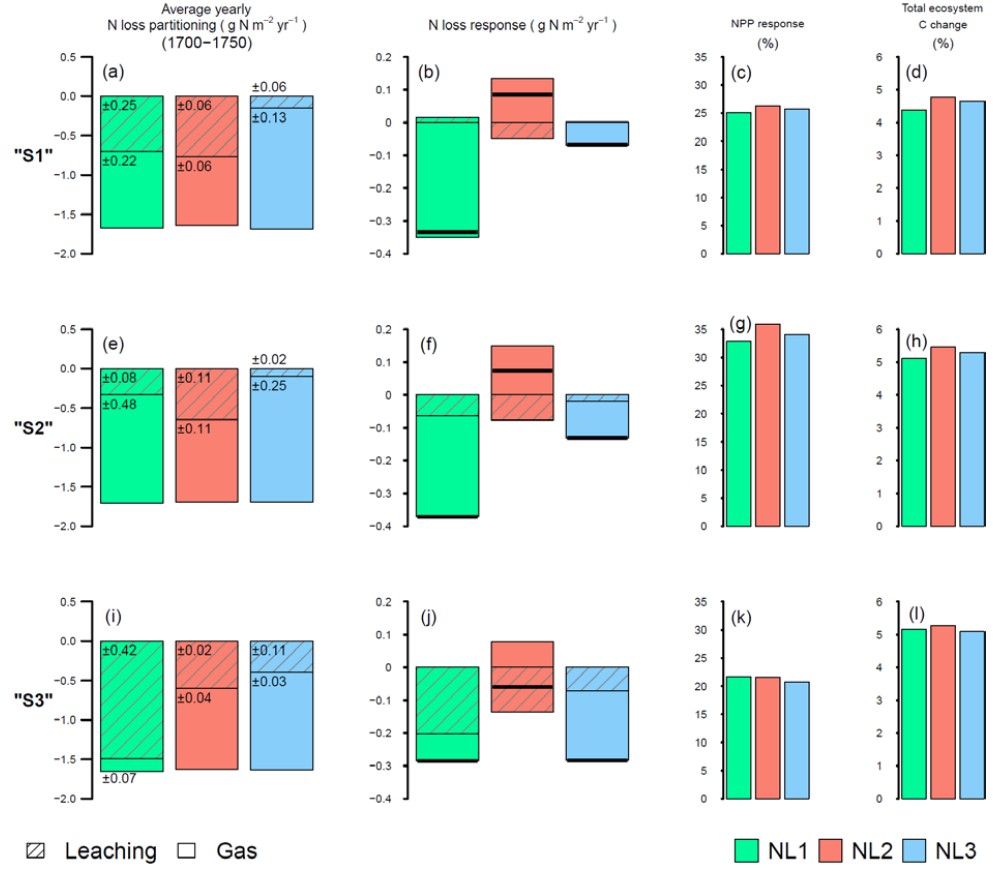

**Figure 2:** Average fate of ecosystem nitrogen (N) input at the three pseudosites "S1", "S2", and "S3" for the 1700-1750 quasi-equilibrium simulation period without perturbation (a, e, i; g N m$^{-2}$ yr$^{-1}$). Average N loss rate responses (g N m$^{-2}$ yr$^{-1}$) to ten years of simulated elevated atmospheric CO$_2$ (eCO$_2$, +200 ppm) (b, f, j). Net primary productivity (NPP) responses (%) after ten years of eCO$_2$, relative to control simulations (c, g, k). Change (%) of total ecosystem carbon (C) after ten years of eCO$_2$, relative to the start of the experiment (d, h, l). Colours indicate the applied N loss algorithm ("NL1", "NL2", "NL3"). For a, b, e, f, i, and j, shading lines indicate leaching, unshaded colour indicates gaseous loss. Numbers on the bars indicate the standard deviation (g N m$^{-2}$ yr$^{-1}$) of the leaching and gas loss components over the 1700-1750 quasi-equilibrium period. For a, e, and i, the sums of N allocation to organic pools (vegetation biomass and soil organic matter) were in the range of -0.07 - 0.01 g N m$^{-2}$ yr$^{-1}$ and were omitted here. External inputs of reactive N (biological N fixation + N deposition) were fixed at 1.631 g N m$^{-2}$ yr$^{-1}$. For b, f, and j, black bars indicate the total N loss responses (gaseous loss + leaching).





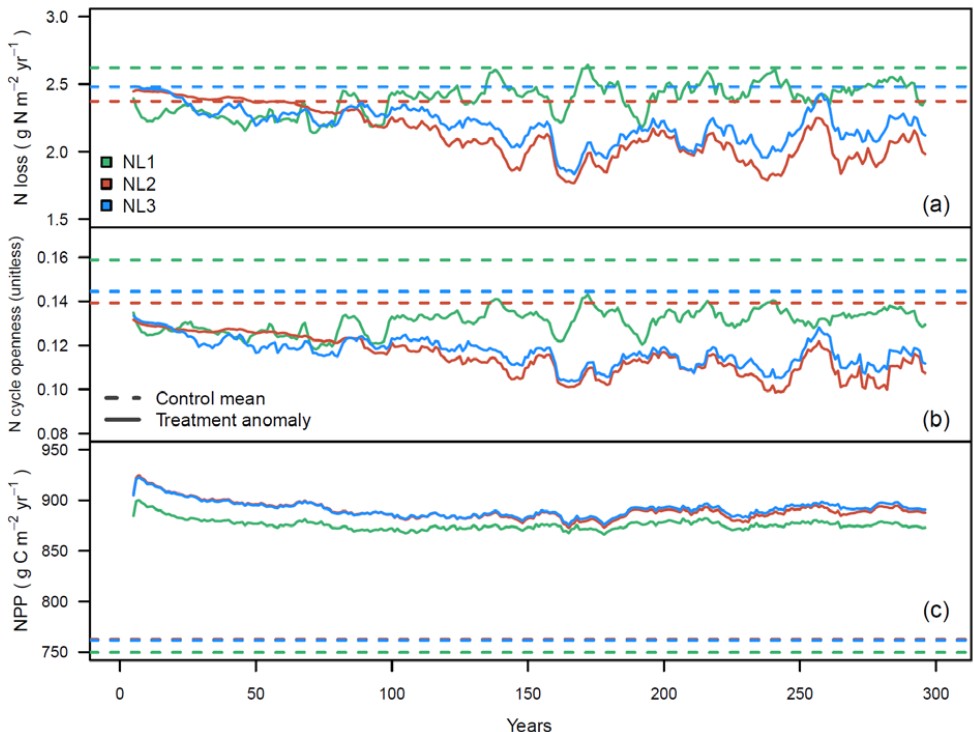

**Figure 3:** Ten year moving average responses of total nitrogen (N) loss (a; g N m$^{-2}$ yr$^{-1}$), N cycle openness (N loss/net N mineralization; b; unitless) and net primary productivity (NPP; c; g C m$^{-2}$ yr$^{-1}$) during the local 300 year eCO$_2$ simulation (380 ppm control + 200 ppm treatment) at the temperate "S1" site using the three N loss models. 300 year mean from the control runs and anomaly (treatment - control + control mean) from treatment simulations.





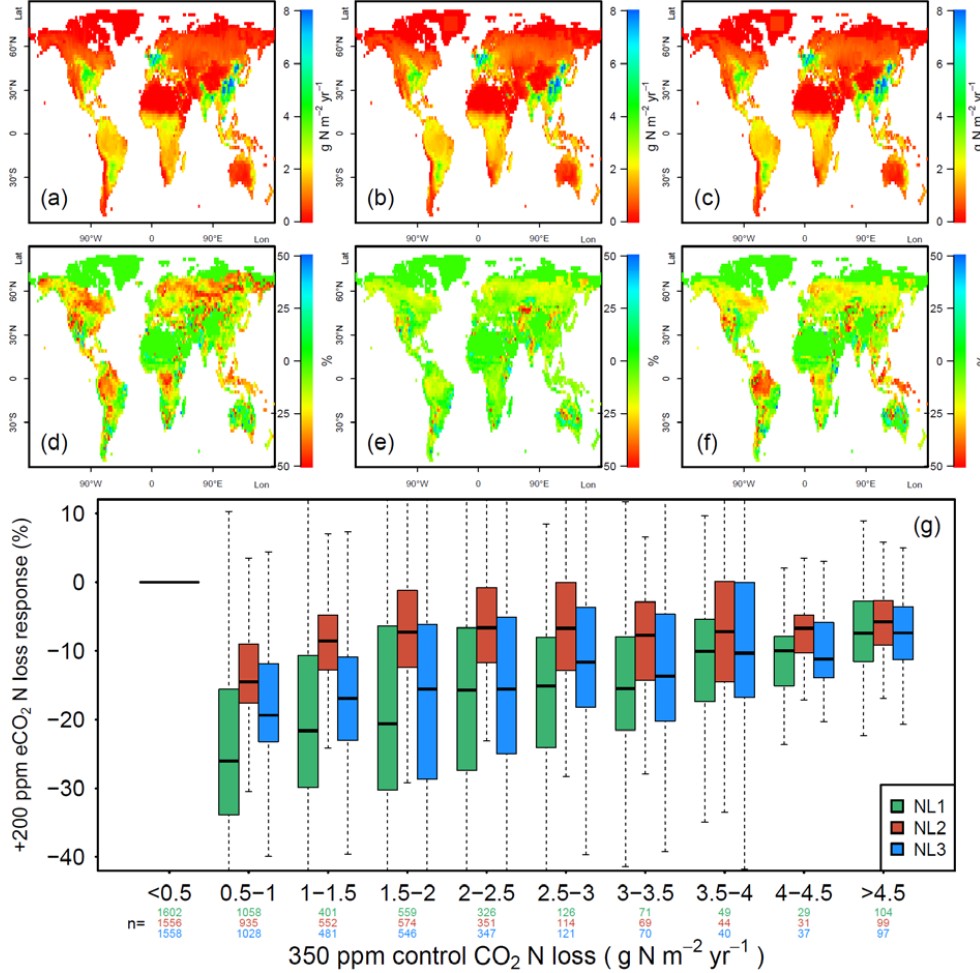

**Figure 4:** (a), (b), (c): Global nitrogen (N) loss rates (g N m$^{-2}$ yr$^{-1}$) for the control state (350 ppm atmospheric CO$_2$ concentration), using the N loss algorithms "NL1" (a), "NL2" (b), and "NL3" (c). (d), (e), (f): Global N loss responses (%) to elevated atmospheric CO$_2$ concentrations (eCO$_2$, +200 ppm gradual increase). (g): Global response ratios (%) plotted against the corresponding control N loss rates, binned in intervals of 0.5 g N m$^{-2}$ yr$^{-1}$. Boxes show median and quartiles, whiskers show the largest/smallest outliers that lie below/above 1.5 times the interquartile range. Further outliers were omitted. Coloured numbers indicate the number of 2° x 2° grid cells that fell into the respective N loss range. The 350-550 ppm difference in atmospheric CO$_2$ concentrations approximately corresponded to the 1988-2052 time span in our global simulations.





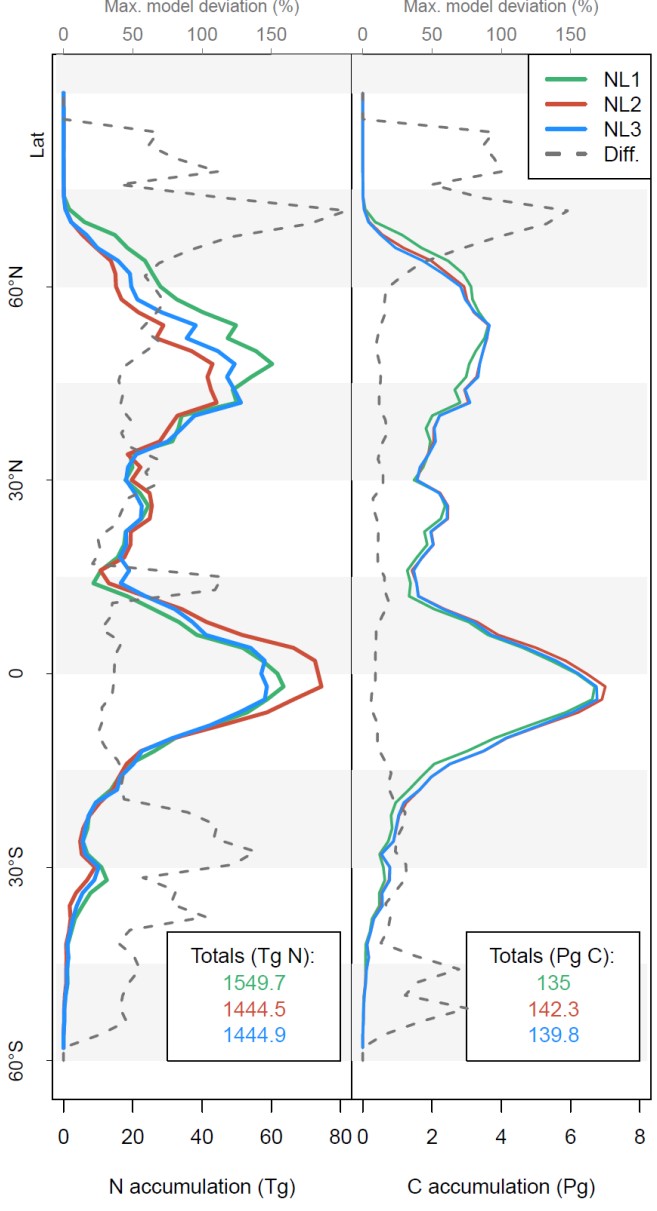

**Figure 5:** Nitrogen (N, left) and carbon (C, right) accumulation during global simulations of elevated atmospheric $CO_2$ concentrations (e$CO_2$) using three different N loss algortihms between 1850 (285 ppm $CO_2$) and 2100 (936 ppm $CO_2$), depicted per 2° latitudinal band and as global totals (coloured numbers). "Diff." shows the respective maximum deviation between the three models as the percentage of the three model mean. The global total values corresponded to average yearly accumulation of: 6.20, 5.78, 5.78 Tg N yr$^{-1}$ ("NL1", "NL2", "NL3") and 0.54, 0.57, 0.56 Pg C yr$^{-1}$.



**Appendix A: Additional figures**

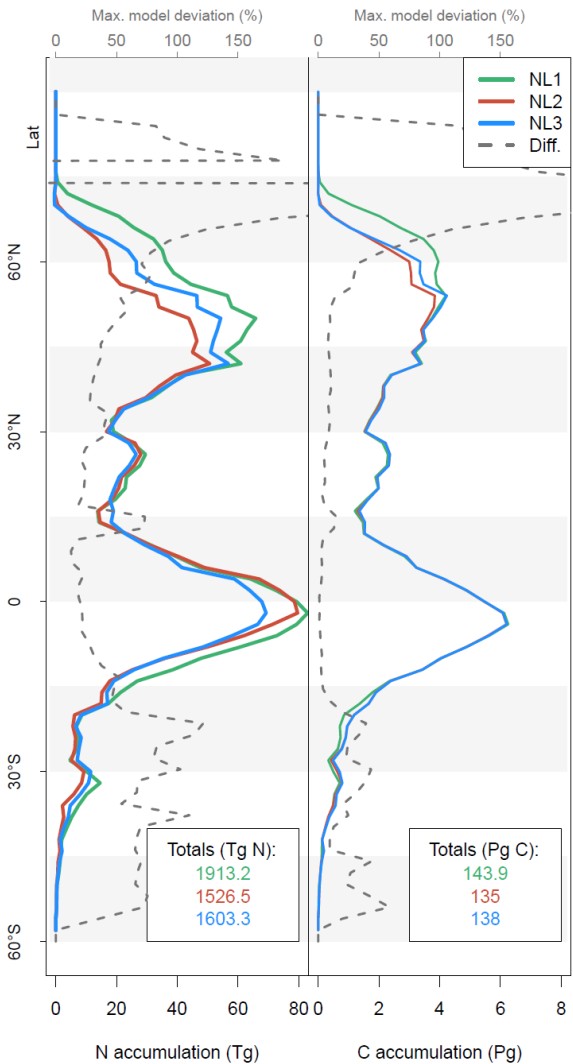

**Figure A1:** Nitrogen (N, left) and carbon (C, right) accumulation during global simulations of elevated atmospheric $CO_2$ concentrations (e$CO_2$) using three different N loss algortihms between 1850 (285 ppm $CO_2$) and 2100 (936 ppm $CO_2$), depicted per 2° latitudinal band and as global totals (coloured numbers). "Diff." shows the respective maximum deviation between the three models as the percentage of the three model mean. The global total values corresponded to average yearly accumulation of: 7.65, 6.11, 6.41 Tg N yr[-1] ("NL1", "NL2", "NL3") and 0.58, 0.54, 0.55 Pg C yr[-1]. The difference to Fig. 5 was the usage of an O-CN terrestrial biosphere model version that employed fixed C:N stoichiometry in all soil and vegetation pools (Meyerholt and Zaehle, 2015).




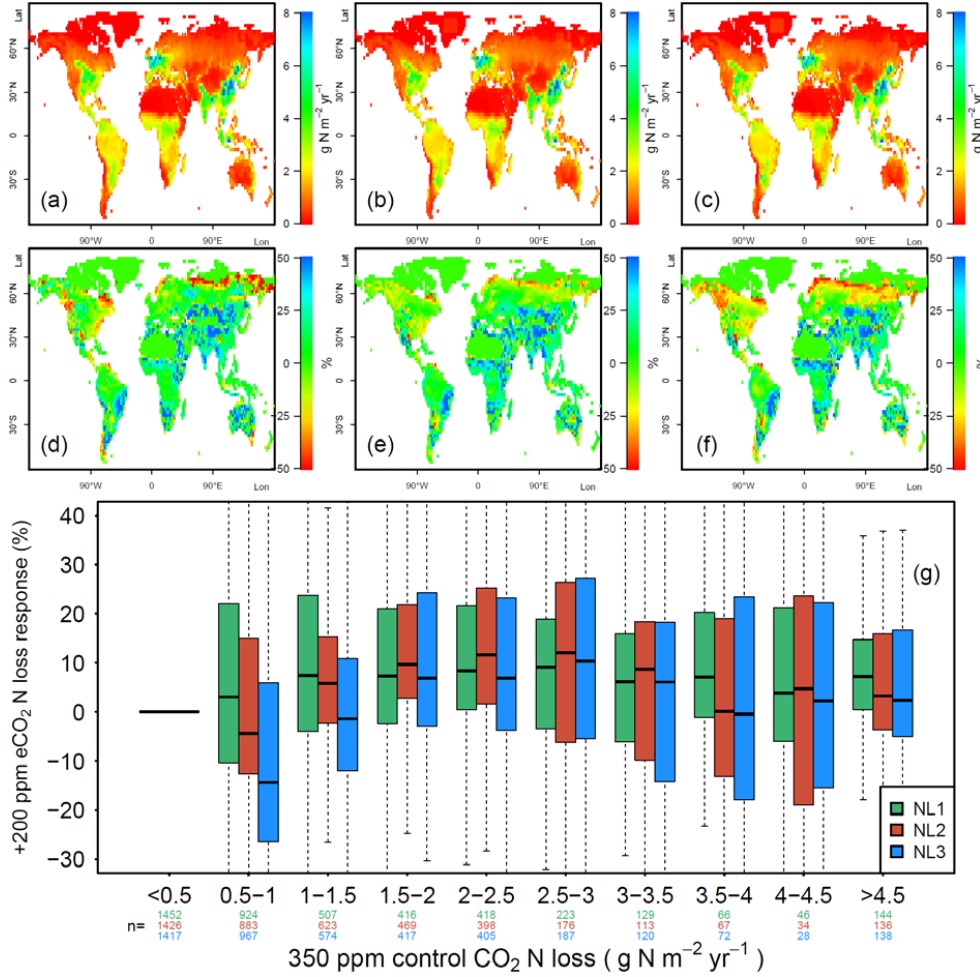

**Figure A2:** (a), (b), (c): Global nitrogen (N) loss rates (g N m$^{-2}$ yr$^{-1}$) for the control runs (350 ppm atmospheric CO$_2$ concentration), using the N loss algorithms "NL1" (a), "NL2" (b), and "NL3" (c). (d), (e), (f): Global N loss responses (%) to elevated atmospheric CO$_2$ concentrations (eCO$_2$, +200 ppm). (g): Global response ratios (%) plotted against the corresponding control N loss rates, binned in intervals of 0.5 g N m$^{-2}$ yr$^{-1}$. Boxes show median and quartiles, whiskers show the largest/smallest outliers that lie below/above 1.5 times the interquartile range. Further outliers were omitted. Coloured numbers indicate the number of 2° x 2° grid cells that fell into the respective N loss range. The 350-550 ppm difference in atmospheric CO$_2$ concentrations approximately corresponded to the 1988 - 2052 time span in our global simulations. The difference to Fig. 4 was that these simulations also included transient climate and N deposition rates according to the RCP8.5 scenario.



## Appendix B: Nitrogen loss algorithm description for NL1

Gaseous nitrogen losses

NL1 includes the adsorption of ammonium ($NH_4^+$) ions to clay particles, following Li et al. (1992). Thereby, the $NH_4^+$ fraction of the soil pool of available inorganic nitrogen (N) is first reduced according to the soil clay content. The anaerobic volume fraction of the soil (*anvf*; Eq. B1) is estimated from the fractional soil moisture content (θ) by an empirical function:

$$anvf = 1 - e^{(\frac{\theta}{0.8})^8} \qquad . \qquad \text{(B1)}$$

In the aerobic part of the soil (1-*anvf*), the fraction of ammonium ($NH_{4,\ aerob}$) is subject to nitrification, with the gross rate ($N_{nit}$; Eq. B2) depending on $NH_4^+$ concentrations, as well as response functions for temperature ($f_{nit}(T)$) and soil pH ($g_{nit}(pH)$):

$$N_{nit} = \alpha * f_{nit}(T) * g_{nit}(pH) * NH_{4,aerob} \quad , \qquad \text{(B2)}$$

where α is chosen such that at 20°C and favourable pH conditions, 10% of the ammonium is nitrified per day.

The temperature function (Eq. B3) is taken from Xu-Ri and Prentice (2008):

$$f_{nit} = (\frac{(70-T_{soil})}{(70-38)})^{12} * e^{12*\frac{(T_{soil}-38)}{(70-38)}} \quad , \qquad \text{(B3)}$$

where $T_{soil}$ is the soil temperature in °C. The soil pH function (Eq. B4) is taken from Zhang et al. (2002):

$$g_{nit} = -0.0604 * pH^2 + 0.7347 * pH - 1.2342 \qquad . \qquad \text{(B4)}$$

N₂O loss from nitrification ($N_2O_{nit}$; Eq. B5) is estimated from the gross nitrification rate (Li et al., 2000):

$$N_2O_{nit} = f_T * 0.0008 * N_{nit} \qquad , \qquad \text{(B5)}$$

where $f_T$ (Eq. B6) is another temperature function:

$$f_T = 2.72^{34.6-\frac{9615}{T_{soil}+273.15}} \qquad . \qquad \text{(B6)}$$

Loss of NO from nitrification (Eq. B7) follows the formulation by Li et al. (2000) with an additional component representing chemonitrification following Kesik et al. (2005):

$$NO_{nit} = (f_T * 0.0025 + f_{Tk} * f_{pHk}) * N_{nit} \quad , \qquad \text{(B7)}$$

with functions representing the temperature (Eq. B8) and pH (Eq. B9) sensitivity of chemonitrification:

$$f_{Tk} = e^{\frac{-31494}{(T_{soil}+273.15)*8.3144}} \quad , \qquad \text{(B8)}$$



$$f_{pHk} = 20 * 16565 * e^{-1.62*pH} \qquad . \tag{B9}$$

Gross denitrification ($N_{denit}$; Eq. B10) of the fraction of $NO_3^-$ under anaerobic conditions ($NO_{3,anaerob}$) is modelled as a function of the $NO_3^-$ concentration, microbial respiration, temperature and pH:

$$N_{denit} = \beta * f_{denit} * g_{denit} * NO_{3,anaerob} * timestep^{-1} \qquad , \tag{B10}$$

where β (Eq. B11) is a function that describes the regulatory influence of soil microbial activity and $NO_{3^-}$ concentrations on gross denitrification (Li et al., 2000):

$$\beta = \frac{R_{mb}}{R_{mb}+K_R} * \frac{NO_3}{NO_3+K_{NO3}} \qquad , \tag{B11}$$

where $R_{mb}$ is the microbial respiration rate, $NO_3$ is the soil $NO_3^-$ concentration (aerobic and anaerobic), and $K_R$ and $K_{NO3}$ are half-saturation constants. The temperature sensitivity of denitrification ($f_{denit}(T)$; ; Eq. B12) is taken

from Xu-Ri and Prentice (2008):

$$f_{denit}(T) = e^{308.56*\left(\frac{1}{68.02} - \frac{1}{(T_{soil}+46.02)}\right)} \qquad . \tag{B12}$$

The sensitivity of denitrification to soil pH is described by $g_{denit}(pH)$ (Eq. B13):

$$g_{denit}(pH) = 1 - \frac{1}{1+e^{\frac{pH-4.25}{0.5}}} \qquad . \tag{B13}$$

Gaseous N losses from denitrification ($NO_{denit}$, Eq. B14; $N_2O_{denit}$, Eq. B15; $N_{2,denit}$, Eq. B16) are then estimated

from the gross denitrification rate, taking into account the different sensitivities to soil pH for the respective proportions of emissions of NO, $N_2O$, and $N_2$ (Li et al., 2000):

$$NO_{denit} = \beta_{NO} * f_{denit} * g_{denit,NO} * N_{denit} \quad , \tag{B14}$$

$$N_2O_{denit} = \beta_{N2O} * f_{denit} * g_{denit,N2O} * N_{denit} \qquad , \tag{B15}$$

$$N_{2,denit} = N_{denit} - NO_{denit} - N_2O_{denit} \qquad , \tag{B16}$$

where $\beta_{NO}$ and $\beta_{N2O}$ are constants and $g_{denit,NO}$ (Eq. B17) and $g_{denit,N2O}$ (Eq. B18) are functions that scale emission of different gaseous N compounds with soil pH:

$$g_{denit,NO} = \frac{g_{denit}}{h_{N2O}*1.825} \qquad , \tag{B17}$$

$$g_{denit,N2O} = \frac{h_{NO}}{h_{N2O}*1.434} \qquad , \tag{B18}$$

with the NO- and $N_2O$-specific functions $h_{NO}$ (Eq. B19) and $h_{N2O}$ (Eq. B20):

$$h_{NO} = 1 - \frac{1}{1+e^{\frac{pH-5.25}{1}}} \qquad , \tag{B19}$$

$$h_{N2O} = 1 - \frac{1}{1+e^{\frac{pH-6.25}{1.5}}} \qquad . \tag{B20}$$





The volatilization of dissolved $NH_4^+$ to $NH_3$ ($NH_{3,vol}$; Eq. B21) depends on soil pH:

$$NH_{3,vol} = 10^{\frac{4.25-pH}{1+10^{4.25-pH}}} * NH_{4,soil} * d_{ox} \qquad , \qquad\qquad\qquad (B21)$$

with the diffusion coefficient $d_{ox}$ that depends on soil moisture and also determines volatilization losses from the soil N pool of other N gases:

$$NO_{vol} = d_{ox} * NO_{soil} \qquad , \qquad\qquad\qquad (B22)$$

$$N_2O_{vol} = d_{ox} * N_2O_{soil} \qquad , \qquad\qquad\qquad (B23)$$

$$N_{2,vol} = d_{ox} * N_{2,soil} \qquad . \qquad\qquad\qquad (B24)$$

Nitrogen leaching

10    Leaching of $NH_4^+$ and $NO_3^-$ occurs in proportion to the water lost from soil drainage, calculated as described by De Rosnay and Polcher (1998).





**Table B1: List of variables and parameters used in the nitrogen (N) loss algorithm "NL1".** $NH_4^+$ = ammonium; $N_2O$ = nitrous oxide; NO = nitric oxide; $NO_3^-$ = nitrate; $NH_3$ = ammonia; $N_2$ = dinitrogen; dt = timestep.

| Variable / Parameter | Description | Value, Unit |
|---|---|---|
| $anvf$ | Anaerobic volume fraction of the soil (Eq. B1) | - |
| $\theta$ | Fractional soil moisture content (Eq. B1) | - |
| $NH_{4,aerob}$ | $NH_4^+$ in the aerobic fraction of the soil N pool (Eq. B2) | g N m$^{-2}$ |
| $N_{nit}$ | Gross N nitrification rate (Eq. B2) | g N m$^{-2}$ dt |
| $\alpha$ | Factor to scale nitrification activity (Eq. B2) | 1.2 |
| $f_{nit}(T)$ | Temperature response function of nitrification (Eq. B2) | - |
| $g_{nit}(pH)$ | pH response function of nitrification (Eq. B2) | - |
| $T_{soil}$ | Soil temperature | °C |
| $N_2O_{nit}$ | $N_2O$ emission from nitrification (Eq. B5) | g N m$^{-2}$ dt |
| $f_T$ | Temperature function for $N_2O$ emission (Eq. B6) | - |
| $NO_{nit}$ | NO emission from nitrification (Eq. B7) | g N m$^{-2}$ dt |
| $f_{Tk}$ | Temperature function for chemonitrification (Eq. B8) | - |
| $f_{pHk}$ | pH function for chemonitrification (Eq. B9) | - |
| $N_{denit}$ | Gross denitrification rate (Eq. B10) | g N m$^{-2}$ dt |
| $\beta$ | Microbe function of gross denitrification (Eq. B11) | - |
| $f_{denit}(T)$ | Temperature function for denitrification (Eq. B12) | - |
| $g_{denit}(pH)$ | pH function for denitrification (Eq. B13) | - |
| $NO_{3,anaerob}$ | $NO_3^-$ in anaerobic fraction of the soil N pool (Eq. B10) | g N m$^{-2}$ |
| $R_{mb}$ | Microbial respiration rate (Eq. B11) | dt |
| $K_R$ | Half-saturation constant (Eq. B11) | dt |
| $K_{NO3}$ | Half-saturation constant (Eq. B11) | g N m$^{-2}$ |
| $NO_{denit}$ | NO loss from denitrification (Eq. B14) | g N m$^{-2}$ dt |
| $N_2O_{denit}$ | $N_2O$ loss from denitrification (Eq. B15) | g N m$^{-2}$ dt |
| $N_{2,denit}$ | $N_2$ loss from denitrification (Eq. B16) | g N m$^{-2}$ dt |
| $\beta_{NO}$ | Constant (Eq. B14) | 0.78 |
| $\beta_{N2O}$ | Constant (Eq. B15) | 0.54 |
| $g_{denit,NO}$ | pH sensitivity function for NO denitrification (Eq. B17) | - |
| $g_{denit,N2O}$ | pH sensitivity function for $N_2O$ denitrification (Eq. B18) | - |
| $h_{NO}$ | pH sensitivity function for NO denitrification (Eq. B19) | - |
| $h_{N2O}$ | pH sensitivity function for $N_2O$ denitrification (Eq. B20) | - |
| $NH_{3,vol}$ | Volatilization of $NH_3$ from the soil N pool (Eq. B21) | g N m$^{-2}$ dt |
| $NH_{4,soil}$ | $NH_4^+$ concentration in the soil N pool (Eq. B21) | g N m$^{-2}$ |
| $d_{ox}$ | Soil moisture dependent diffusion coefficient (Eq. B22) | 0.001-0.005 dt |
| $NO_{vol}$ | Volatilization of NO from the soil N pool (Eq. B22) | g N m$^{-2}$ dt |
| $N_2O_{vol}$ | Volatilization of $N_2O$ from the soil N pool (Eq. B23) | g N m$^{-2}$ dt |
| $N_{2,vol}$ | Volatilization of $N_2$ from the soil N pool (Eq. B24) | g N m$^{-2}$ dt |
| $NO_{soil}$ | NO concentration in the soil N pool (Eq. B22) | g N m$^{-2}$ |
| $N_2O_{soil}$ | $N_2O$ concentration in the soil N pool (Eq. B23) | g N m$^{-2}$ |
| $N_{2,soil}$ | $N_2$ concentration in the soil N pool (Eq. B24) | g N m$^{-2}$ |



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
