# Peer review of "Controls of terrestrial ecosystem nitrogen loss on simulated productivity responses to elevated CO2"

_Biogeosciences, 2018_

## Referee Comment (RC1) · Anonymous Referee #1 · 2 Jul 2018

This was a well performed study, and I recommend it for publication after some minor clarifications. It is an important study within the global C and N cycle Community.

One clarification that is needed, is how agriculture is treated since the authors report and draw conclusions about the incluence in crop dominated areas, but to my knowledge crops are not modelled in O-CN.

The paper would benifit from a Conclusion section, where the authors summarize the good conclusions that they draw from this study.

It was hard to follow the "pseudo sites" ($S_1$,$S_2$,$S_3$), better to name them: $S_0$, $S_T$ and $S_P$.

---

## Referee Comment (RC2) · Anonymous Referee #2 · 5 Jul 2018

This study investigates the influences of three different N loss mechanisms in ecosystem models on carbon cycle under elevated CO2. The topic is of general interest and the simulation experiments are well designed, and I recommend it for publication after minor revision.

The result that N loss showed minor impact on carbon sequestration is unexpected, in particular in northern temperate and high-latitude regions where ecosystems are usually thought to be strongly N-limited. The authors did not give exact reasons for this phenomenon but inferred that the varying C:N ratio in the O-CN model is a possible explanation, although it cannot fully explain the result. Here, in my opinion, it will be

more persuasive to simply give the changes in plant N uptake which directly controls vegetation productivity.

Other comments: Line 28 Page 10: change "N available" to "N availability"

---

## Referee Comment (RC3) · Anonymous Referee #3 · 16 Jul 2018

Overall, I really like the paper and I think it is worthwhile to publish. The paper highlights the different N loss routines in DVMs, and how this affects model performance in both N loss and C gain. Especially the experimental setup of running the model makes it useful and applicable for others. Also, the paper is well written and has good graphics.

However, I have some questions about the paper. First, while the N loss differences of the experiments are well explained and show with nice graphics, the C accumulation is more difficult to understand. Especially in the global simulation, the three different routines show quite different N accumulation (fig 5), while the C accumulation seems to be insensitive to the N accumulation. This is the same when the C:N ratio is constant

rather than flexible. The authors state that 'the exact mechanisms are difficult to discern', but that leaves me puzzled.. Did the authors look into more differences besides the C:N ratio's? How does this result link to the earlier 'experiments' in the paper? Where is most of the N that is accumulated stored and how does this relate to the C accumulation?

I have a few more questions and comments which I will go through one by one:

P1, line 37: the authors refer to figure 1, but this is confusing in this part of the paper. P4-5 & figure 1: the N loss formulations are well explained in words, but figure 1 is difficult to read on its own. Also, would it be possible to add N uptake somewhere in the methods? Since later in the paper we look at both N loss and C gain, it would be good to know the general N update scheme of O-CN, and how the Nloss routine of NL3 is altering the overall Nuptake routine in that formulation?

P4, line 17: is leaching of $NH_4^+$ equal to leaching of $NO_3^-$?

P6, line 27: just to be sure, in the global model run you use fertilizer application, but no N-deposition? What is the rationale?

P8, paragraph 3.1.3: This paragraph could use more explanation. Especially figure 2j is still unclear to me, as an example: why, with $eCO_2$, is the leaching loss so much reduced? Is this because N uptake in mainly inorganic N and will happen before leaching? Why is the gaseous loss in NL2 so much reduced?

P10. Line 13: relative to 1850 values (285 ppm)?

P10, line 34: The 1 sentence for figure 4g is quite limited. Could this be extended? It is relative to control $CO_2$ N loss? And how does it stand for N limitation?

---

## Author Response (AR1)

We are very grateful to Anonymous Referee #1 for the positive comments on our manuscript and constructive suggestions to further improve its quality.

**One clarification that is needed, is how agriculture is treated since the authors report and draw conclusions about the incluence in crop dominated areas, but to my knowledge crops are not modelled in O-CN.**

We did not include crops (and the associated fertilizer use) in the local experiments, but do so in the global simulations, which is where we refer to N loss hotspots in regions with high density of agricultural land use (p10 l11). While this is in principle described in the Methods section, we will emphasize this difference stronger in the revised MS.

*We added a sentence at p6l29 to stronger highlight this difference.*

**The paper would benifit from a Conclusion section, where the authors summarize the good conclusions that they draw from this study.**

We agree and will revise the MS to have seperate sections for our discussion and conclusions.

*We reworked the end of the original discussion section to be the new conclusion section (p14l10).*

**It was hard to follow the "pseudo sites" ($S_1$, $S_2$, $S_3$), better to name them: $S_0$, $S_T$ and $S_P$ .**

We agree and will adapt this more descriptive naming convention.

*Done throughout the manuscript, including Fig. 2.*

**Author response to**

We are very grateful to Anonymous Referee #2 for the positive comments on our manuscript and constructive suggestions to further improve its quality.

**The result that N loss showed minor impact on carbon sequestration is unexpected, in particular in northern temperate and high-latitude regions where ecosystems are usually thought to be strongly N-limited. The authors did not give exact reasons for this phenomenon but inferred that the varying C:N ratio in the O-CN model is a possible explanation, although it cannot fully explain the result. Here, in my opinion, it will be more persuasive to simply give the changes in plant N uptake which directly controls vegetation productivity.**

Following the Referee's advice, we will conduct an additional analysis that tracks the evolution of vegetation N uptake in our simulations. This will most likely result in another supplementary figure that illustrates model $eCO_2$ responses in vegetation N uptake that correspond to the small differences in productivity responses.

*We did not follow up on this, as we provide the appropriate explanation for the phenomenon in question in response to the comments by Referee #3: In addition to varying stoichiometry, differences in total C accumulation occur through model differences in above-/below-ground allocation of accumulated N.*

**Other comments: Line 28 Page 10: change "N available" to "N availability"**

Agreed.

*Done (p10l29).*

**Author response to**

We are very grateful to Anonymous Referee #3 for the positive comments on our manuscript and constructive suggestions to further improve its quality.

**However, I have some questions about the paper. First, while the N loss differences of the experiments are well explained and show with nice graphics, the C accumulation is more difficult to understand. Especially in the global simulation, the three different routines show quite different N accumulation (fig 5), while the C accumulation seems to be insensitive to the N accumulation. This is the same when the C:N ratio is constant rather than flexible. The authors state that 'the exact mechanisms are difficult to discern', but that leaves me puzzled.. Did the authors look into more differences besides the C:N ratio's? How does this result link to the earlier 'experiments' in the paper? Where is most of the N that is accumulated stored and how does this relate to the C accumulation?**

This is indeed a crucial point in this MS. While further investigating our results for better explanation of the observed phenomenon, we found that Figure 5 erroneously displayed the N and C accumulation for 1850-2005 and not for 1850-2100 as claimed. Correcting for this (see revised Figure 5 below) affects the absolute magnitudes of accumulation, but does not change the qualitative observation since differences in global C accumulation are still rather insensitive to differences in N accumulation (especially in the boreal zone), and this phenomenon is not satisfyingly explained by stoichiometric flexibility (Fig. 5 flexible stoich.; Fig. A1 fixed stoich.). As Referee #3 suspects, the underlying reason for the lack of C accumulation response is the partitioning of N accumulation between vegetation and soils. While the initial response to added N would be increased production and vegetation N storage, at the time scale of the analysis N predominantly ends up in soil organic matter because of the longer time scale of soil carbon turnover compared to the relatively quick turnover of vegetation tissue pools.

Figure 5:

[Figure]

Figure A1:

[Figure]

The NL1 model stores relatively more N in the soil, because the soil inorganic pool is more depleted of N in comparison with NL2/3, therefore N uptake and the vegetation N fraction are lower. NL2/3 can store relatively more N in high C:N vegetation, therefore higher N accumulation in NL1 does not result in higher total ecosystem C accumulation, and the model differences in N accumulation are rather supressed in total C accumulation.

The detailed mechanics of this are best illustrated using the fixed stoichiometry version of O-CN, since ecosystem C:N only changes through changes in allocation to pools with different C:N ratios, whereas the flexible stoichiometry version also involves shifts in pool C:N ratios, which makes the explanation more complex and less tractable. Full analysis would also involve regional differences from climate and PFT specifics. However, the flexible stoich. version was found to give more realistic results when predicting ecosystem responses to perturbation (Meyerholt & Zaehle 2015, New Phyt).

We will revise the results and discussion sections with respect to the global analysis to give better explanation of the observed phenomena involving the points above, while keeping the MS reasonably easy to follow and relevant for a broad audience.

Partitioning of N (left) and C (right) accumulation:

[Figure]

Vegetation fraction of accumulated N (left) and C (right)

[Figure]

*We fixed all "latitudinal" plots (Fig. 5, A1-3) to now cover the correct time-span of 1850-2100. We revised the Results and Discussion sections to include the explanation for the low sensitivity of C accumulation to model differences in N accumulation by model differences in N allocation as a consequence of different N loss algorithms (p11l17ff,p12l1ff).*

**P1, line 37: the authors refer to figure 1, but this is confusing in this part of the paper.**

Figure 1 illustrates model approaches to represent ecosystem-level N loss, whereas the cited text more refers to reality. We will remove the reference to Figure 1.

*We removed the reference to Figure 1.*

**P4-5 & figure 1: the N loss formulations are well explained in words, but figure 1 is difficult to read on its own. Also, would it be possible to add N uptake somewhere in the methods? Since later in the paper we look at both N loss and C gain, it would be good to know the general N update scheme of O-CN, and how the Nloss routine of NL3 is altering the overall Nuptake routine in that formulation?**

Following the Referee's advice, we will improve Figure 1 and its caption to be more self-explanatory, and additionally include N uptake by vegetation, as well as litterfall from vegetation to soil organic matter.

Zaehle & Friend (2010) offers a supplementary document that features a detailed description of the O-CN N uptake algorithm. The direct link between N loss and N uptake is soil mineral N availability. For clarity, we will add the O-CN description of N uptake to this MS.

*Figure 1 was extended to depict the entire simulated internal N cycle for better context with the shown N loss algorithms. We also added a seperate description of O-CN N uptake to the Appendix (C), including the interaction with plant N status (p34).*

**P4, line 17: is leaching of NH4+ equal to leaching of NO3-?**

For NL1, leaching occured in proportion to drainage in equal proportions for NH4+ and NO3-. However, a fraction of NH4+ is assumed to be sorbed to clay minerals and thus prevented from leaching and biogeochemical processing. For NL2 and NL3, we simulated one generic inorganic N species to conform with the imitated N loss algorithms.

**P6, line 27: just to be sure, in the global model run you use fertilizer application, but no N-deposition? What is the rationale?**

The global simulations feature constant fertilizer application and N deposition. Thereby, N deposition is not zero, but we avoided it being a perturbing factor over time.

**P8, paragraph 3.1.3: This paragraph could use more explanation. Especially figure 2j is still unclear to me, as an example: why, with eCO2, is the leaching loss so much reduced? Is this because N uptake in mainly inorganic N and will happen before leaching? Why is the gaseous loss in NL2 so much reduced?**

eCO2 drives down soil N concentrations, therefore it drives down leaching. Gas losses in NL2 are increased, not decreased. Gas loss in NL3 is decreased because there is much less excess N. Increased N uptake under eCO2 decreases N soil N concentrations, therefore reducing losses. The exception, as explained in the text, is NL2 gas loss, where increased soil C:N leads to increased N mineralization, which in this case mainly determines gaseous loss.

**P10. Line 13: relative to 1850 values (285 ppm)?**

Certainly, but also relative to 350 ppm (1988), which is the "control state" shown in Figure 4a,b,c. We will formulate this more concisely.

*The formulation is now more concise (p10l14).*

**P10, line 34: The 1 sentence for figure 4g is quite limited. Could this be extended? It is relative to control CO2 N loss? And how does it stand for N limitation?**

We will point out stronger in the caption that this picture refers to the 350-550 ppm (1988-2052) response, and will dedicate additional main text to the observation that N loss responses differ most between models when soil inorganic N availability is low, i.e. high N limitation.

*We added to this paragraph (p10l35) by including some information from the previous paragraph, as they refer to the same phenomenon (high sensitivity of NL1 loss to soil inorganic N availability). From main text and caption, it should be clear that Fig. 4g shows N loss responses relative to control soil N, which here approximately corresponds to degree of N limitation.*

[revised manuscript text omitted]

---

## Author Response (AR2)

**bg-2018-232**

**"Controls of terrestrial ecosystem nitrogen loss on simulated productivity responses to elevated CO2" by Johannes Meyerholt and Sönke Zaehle**

**Author response to**

**Associate Editor Decision: Publish subject to technical corrections (07 Sep 2018)**

*Dear Associate Editor,*

*Thank you for accepting our manuscript subject to technical corrections. We have implemented the corrections as outlined here and in the attached marked-up manuscript version:*

1) May be you wish to mention weathering as an additional source of reactive nitrogen in the abstract and introduction

*We have added weathering as an external source of nitrogen on p 1 and 2 as well as the caption to Figure 1.*

2) First sentence on top of page 8: I do not understand this sentence and the link to SOM C:N

*To avoid confusion, we simplified the paragraph and removed the reference to SOM C:N (p 8).*

3) Make sure that labels (negative signis) are clearly visible in Fig. 4d-f

*This was done.*

4) Please revise your color scale in the different figures to make the figures easily accessible for readers with color blindness

*This was done.*

[revised manuscript text omitted]